# Adaptive Laboratory Evolution of a Microbial Consortium Enhancing Non-Protein Nitrogen Assimilation for Feed Protein Production

**DOI:** 10.3390/microorganisms13061416

**Published:** 2025-06-18

**Authors:** Yi He, Shilei Wang, Yifan Mi, Mengyu Liu, Huimin Ren, Zhengxiang Guo, Zhen Chen, Yafan Cai, Jingliang Xu, Dong Liu, Chenjie Zhu, Zhi Wang, Hanjie Ying

**Affiliations:** 1School of Chemical Engineering, Zhengzhou University, Zhengzhou 450001, China; 18213142823@163.com (Y.H.); shileiwang@zzu.edu.cn (S.W.); miyifan1998@163.com (Y.M.); liumengyu0037@163.com (M.L.); 17326266960@163.com (H.R.); gzx15238071996@163.com (Z.G.); caiyafan@zzu.edu.cn (Y.C.); xujl@zzu.edu.cn (J.X.); 2State Key Laboratory of Biobased Transport Fuel Technology, Zhengzhou University, Zhengzhou 450001, China; 3Henan Key Laboratory of Tea Plant Biology, College of Life Science, Xinyang Normal University, Xinyang 464000, China; chenzhen@xynu.edu.cn; 4National Engineering Research Center for Biotechnology, Nanjing Tech University, Nanjing 211816, China; liudong@njtech.edu.cn (D.L.); zhucj@njtech.edu.cn (C.Z.); yinghanjie@njtech.edu.cn (H.Y.)

**Keywords:** microbial consortium, adaptive laboratory evolution, wheat straw, non-protein nitrogen assimilation, solid-state fermentation

## Abstract

The increasing global demand for protein underscores the necessity for sustainable alternatives to soybean-based animal feed, which poses a challenge to human food security. Thus, the search for sustainable, alternative protein sources is transforming the feed industry in its effort to sustainable operations. In this study, a microbial consortium was subjected to adaptive laboratory evolution using non-protein nitrogen (NPN) and wheat straw as the sole carbon source. The evolved microbial consortium was subsequently utilized to perform solid-state fermentation on wheat straw and NPN to produce feed protein. After 20 generations, the microbial consortium demonstrated tolerance to 5 g/L NPN, including ammonium sulfate, ammonium chloride, and urea, which represents a fivefold increase compared to the original microbial consortium. Among the three NPNs tested, the evolved microbial consortium exhibited optimal growth performance with ammonium sulfate. Subsequently, the evolved microbial consortium was employed for the solid-state fermentation (SSF) of wheat straw, and the fermentation conditions were optimized. It was found that the true protein content of wheat straw could be increased from 2.74% to 10.42% under specific conditions: ammoniated wheat straw (15% *w*/*w*), non-sterilization of the substrate, an inoculation amount of 15% (*v*/*w*), nitrogen addition amount of 0.5% (*w*/*w*), an initial moisture content of 70%, a fermentation temperature of 30 °C, and a fermentation duration of 10 days. Finally, the SSF process for wheat straw was successfully scaled up from 0.04 to 2.5 kg, resulting in an increased true protein content of 9.84%. This study provides a promising approach for the production of feed protein from straw and NPN through microbial fermentation, addressing protein resource shortages in animal feed and improving the value of waste straw.

## 1. Introduction

With the continuous growth of the global population, there has been a persistent increase in the demand for high-quality animal proteins, including meat, dairy products, and eggs [1]. However, the sustainable supply of feed protein for the livestock sector encounters substantial challenges, as the limited availability of arable land is insufficient to produce adequate plant-derived feed proteins, such as soybean meal, in numerous regions [2]. For example, in China, more than half of the soybean meal is imported [3]. Obviously, it is imperative to investigate novel feed protein sources to replace plant-derived feed proteins. The recycling and utilization of agricultural and industrial wastes have emerged as a promising strategy to mitigate or resolve the issue of feed protein shortages in the livestock industry. However, the low protein content, presence of anti-nutritional factors, and poor digestibility of the waste materials (e.g., straw, distillers’ grains, and oilseed meals) constrain their effectiveness as substitutes for plant-based protein [4,5,6].

Various physical, chemical, and biological methods have been investigated to enhance the inclusion levels of agricultural and industrial waste in animal feed [7,8,9]. Currently, physical treatments, including grinding, extrusion, and steam explosion, primarily focus on disrupting the lignocellulosic structure of such waste materials, such as straw, thereby improving their digestibility and accessibility for subsequent processing [10,11,12]. Chemical treatments (e.g., acid/alkali hydrolysis or oxidation) are further employed to decompose the complex carbohydrates into monosaccharides or disaccharides and to partially remove lignin [13,14,15]. Compared to physical methods, chemical treatments can enhance the nutritional value of the substrate [16]; however, they may generate by-products that are harmful to animal growth and cause environmental pollution [17,18]. Conventional biological fermentation techniques, such as those utilizing cellulolytic fungi or bacteria, predominantly aim to degrade fibrous components to improve the bioavailability of raw materials [19]. Numerous studies have reported that solid-state fermentation (SSF) of agricultural residues, like straw, can only improve the relative protein content due to substrate mass loss, without improving the absolute protein content. More critically, these methods predominantly focus on enhancing the intrinsic protein content of the raw material, rather than generating new protein through microbial biosynthesis. Consequently, these studies do not achieve a net increase in total protein content and fail to significantly reduce reliance on plant-derived protein sources in animal feed.

The successful augmentation of protein content in agricultural waste is contingent upon the ability of microorganisms to concurrently metabolize carbon sources and convert non-protein nitrogen (NPN) in the substrate into microbial protein [20]. It is only through microbial transformation processes that the absolute protein content can be increased, thereby enabling the effective substitution of traditional protein sources. For waste materials with complex components, such as straw, a single microbial strain may not efficiently degrade cellulose, hemicellulose, and lignin, or effectively assimilate NPN for protein synthesis [21,22]. In addition, individual microorganisms often encounter challenges, such as poor environmental adaptability, limited substrate utilization, and low efficiency [23,24,25]. Microbial consortia exhibit enhanced robustness in response to substrate variations and environmental fluctuations [26,27]. Through the division of labor among different species, mixed strains alleviate the metabolic burden on individual strains, thereby optimizing overall energy utilization and protein synthesis efficiency [28,29]. The primary methods for obtaining microbial consortia include synthetic assembly, collection of natural microbiomes, and targeted domestication [30,31]. Synthetic microbial consortia have demonstrated excellent degradation capabilities. For example, the researchers have employed multi-omics approaches to construct microbial consortia, utilizing ammonium sulfate and straw as nitrogen and carbon sources in the SSF of corn straw. The findings indicated a 42.08% degradation of corn straw, with a 1.05-fold increase in protein content within the substrate [32]. However, synthetic consortia are inherently complex and susceptible to instability and efficiency loss under variable environment conditions. In contrast, natural microbial consortia offer greater stability and adaptability, although their functional efficiency may be compromised by the presence of inactive or redundant species [33].

Adaptive laboratory evolution (ALE) represents an innovative technique in the field of biology that aligns with the objectives of targeted domestication and has been widely applied to various microorganisms [34]. The method operates by gradually increasing environmental stress during cultivation and periodically selecting for high-performance evolved strains with enhanced traits, such as acid tolerance, thermal tolerance, and toxin resistance [35,36]. In contrast to metabolic engineering and traditional mutagenesis, ALE enables the spontaneous accumulation of beneficial mutations while maintaining a relatively low rate of deleterious mutations. This approach offers an effective strategy for strain improvement without requiring a detailed understanding of the complex mechanisms underlying cellular metabolic networks [37,38,39,40]. For instance, a microbial consortium enriched from compost was thermally adapted and then applied for straw degradation. Under conditions of 55 °C, the evolved consortium decomposed 78.21–81.73% of straw within 7 days [41]. In another study, a microbial community extracted from natural environments was subjected to directed ALE, resulting in a significant improvement in lignocellulose conversion efficiency, with the cellulose degradation rate increasing from 84.4% to 92.1% [40]. Furthermore, the researchers enriched a native microbial community by using ibuprofen as the sole carbon source and conducted ALE to enhance ibuprofen degradation, resulting in the ibuprofen mineralizing community C6 [42].

In this study, a soil microbial consortium was evolved through ALE to enhance its ability to utilize NPN with wheat straw as the sole carbon source. The evolved microbial consortium was subsequently employed to produce feed protein from wheat straw. The NPN tolerance and assimilation capabilities of the microbial consortium were evaluated both prior to and following ALE. Furthermore, the consortium’s ability to perform the solid-state fermentation of wheat straw for protein production was assessed. This study provides a sustainable approach to valorizing agricultural residues into high-protein feed, thereby addressing both the resource scarcity and environmental challenges associated with conventional feed production.

## 2. Materials and Methods

### 2.1. Strains, Wheat Straw, and Medium Used in the Experiment

Five samples of rhizospheric soil were collected from various wheat fields in the vicinity of Xingyang Town, Zhengzhou City, on 5 April 2023. Following homogenization and impurity removal via sieving, the soil samples were stored at 4 °C in a refrigerator.

Wheat straw was collected from a local farm surrounding Zhengdong New District (Zhengzhou, China) in the autumn of 2023. The wheat straw was processed using a high-speed comminutor to crush and then sieve using a sieve with a particle size of 40 mesh. It was dried at 65 °C to a constant weight and was stored in a self-sealing bag until experimental usage. The main components of wheat straw were cellulose, hemicellulose, and lignin, accounting for 28.88 ± 0.03% total solids (TSs), 4.90 ± 0.04% TSs, and 12.95 ± 1.11% TSs, respectively.

Approximately, 1 g of soil sample was added into 100 mL of sterile saline (AR, Aladdin), incubated in a shaking incubator (200 rpm, 30 °C) for 30 min, and settled naturally for 5 min. Then the supernatant was added to 100 mL of nutrient-enriched medium at an inoculation amount of 1% and incubated by shaking (150 rpm, 30 °C) for 24 h to obtain the seeds of the original microbial consortium. Unless otherwise noted, the pH of all the experiments was the natural-condition pH.

Nutrient-enriched medium: 5 g/L tryptone (BR, OXOID), 20 g/L glucose (BR, OXOID), 2.44 g/L Na_2_HPO_4_ (AR, Aladdin), 1.52 g/L KH_2_PO_4_ (AR, Aladdin), 2 mg/L FeSO_4_·7H_2_O (AR, Aladdin), 0.2 g/L MgSO_4_·7H_2_O (AR, Aladdin), 2 g/L yeast powder (BR, OXOID), 0.05 g/L CaCl_2_·2H_2_O (AR, MACKLIN), and 5 mg/L EDTA-2Na (AR, Solarbio).

Evolution medium: 50 g/L wheat straw, 1.52 g/L KH_2_PO_4_ (AR, Aladdin), 2.44 g/L Na_2_HPO_4_ (AR, Aladdin), 0.05 g/L CaCl_2_·2H_2_O (AR, MACKLIN), 0.2 g/L MgSO_4_·7H_2_O (AR, Aladdin), 2 mg/L FeSO_4_·7H_2_O (AR, Aladdin), and 5 mg/L EDTA-2Na (AR, Solarbio).

### 2.2. Adaptive Laboratory Evolution of the Microbial Consortium

The original microbial consortium was inoculated into conical flasks containing 100 mL of evolution medium with urea (AR, Solarbio), ammonium chloride (AR, Solarbio), and ammonium sulfate (AR, Aladdin) as the sole nitrogen source. These flasks were then placed in a constant-temperature shaker at 30 °C with a shaking speed of 150 rpm for aerobic cultivation. After 48 h of incubation, the growth condition of the microbial consortium and its ability to utilize NPN were assessed. Subsequently, 1% of the culture fluid was used as an inoculum and transferred to fresh modified medium for continued cultivation under constant conditions. During the evolution process, once the nitrogen source utilization ability and growth state of the microbial consortium appeared to stabilize, the concentration of NPN in the medium was gradually increased. The concentrations of NPN (urea, ammonium chloride, and ammonium sulfate) at different generations during the ALE of the microbial consortium are presented in Appendix A. This experimental evolution was carried out for 20 cycles and lasted 60 days. The growth of the bacterial culture was monitored by measuring the optical density (OD_600_) of the bacterial solution at 600 nm using a UV–Vis spectrophotometer (Cary 60 UV-Vis, Agilent Technologies Inc., Santa Clara, CA, USA), and part of the bacterial solution was centrifugally separated by centrifuge (5810R, Eppendorf AG, Hamburg, Germany) at 8000 rpm (7703× *g*) for 10 min to obtain the supernatant. The Kjeldahl method was used to determine the nitrogen content in three parallel samples [43].

### 2.3. SSF of Wheat Straw by the Microbial Consortium

For the preparation of fermentation, the original microbial consortium and evolved microbial consortium (20th generation) were inoculated into the corresponding culture medium and cultured. A total of 500 μL of the original microbial consortium bacterial solution was pipetted from the bacteria-preserving tube and inoculated into 100 mL of sterile nutrient-enriched medium. The culture was incubated at 30 °C in a shaking incubator at 150 rpm for 48 h, after which the activated bacterial suspension was obtained and used for subsequent experiments. A total of 500 μL of the evolved microbial consortium bacterial solution was inoculated into 100 mL sterile evolved medium containing 2 g/L NPN (0.94 g ammonium sulfate) at 30 °C in a shaking incubator for 72 h at 150 rpm. After incubation, the upper layer of the bacterial liquid was left to stand for 30 min of resting and was reserved for use.

SSFs were carried out in 500 mL triangular flasks. A total of 40 g of wheat straw dried to a constant weight was placed in 500 mL triangular flasks, and the fermentation moisture content was adjusted to 65% (*v*/*w*) with an additional 0.5% (*w*/*w*) nitrogen source (ammonium sulfate), then sterilized in an autoclave for 20 min at 115 °C. Inoculum of microbial consortium was added at the fermentation substrate at a 5% (*v*/*w*) inoculation amount. After mixing well, the fermentation system was placed in a constant-temperature incubator at 30 °C for fermentation. The experiment was conducted in three parallel experiments. Sampling was performed only once in a reactor to prevent the effect of volume changes. Fermentation samples were collected at 0, 5, 10, 15, 20, and 30 days to analyze the fermentation performance, including crude protein and true protein content.

#### 2.3.1. Optimization of Fermentation Conditions

A series of studies were conducted to optimize the conditions for SSF of wheat straw. The variables tested included pretreatment of wheat straw (ammonia solution, NaOH), fermentation time (0, 5, 10, 15, 20 days), inoculation amount (2%, 5%, 10%, 15%, 20%, 25% *v*/*w*), nitrogen source addition (0.25%, 0.5%, 1.0%, 1.5%, 2.0%, 2.5% *w*/*w*), initial moisture content (55%, 60%, 65%, 70%, 75%, 80%), temperature (20 °C, 25 °C, 30 °C, 35 °C, 40 °C, 45 °C), and sterilization conditions (sterilization and non-sterilization). The experiment was carried out in three parallel experiments with only one sampling in one reactor to prevent the effect of volume changes.

#### 2.3.2. Pretreatment of Wheat Straw

The wheat straw was crushed through a 40-mesh (0.425 mm) sieve and dried, and the moisture content of the wheat straw was adjusted to 30%, which was mixed well and placed in an airtight container. Then, ammonia solution (0–15–20–28%) was added to the bottom of the raw material at 35 °C in the incubator for ammonia treatment for 9 days. At the end of the incubation, it was dried at 65 °C for drying and preservation.

The dried wheat straw was crushed through a 40-mesh sieve (0.425 mm). Then, it was mixed evenly with NaOH solution at ratios of 1:1 (0%, 2%, 4%, 6%, and 8%). The mixture was placed into conical flasks, pressed tightly, and filled with a certain amount of CO_2_. Subsequently, the flasks were incubated at room temperature for 10 days. After the treatment was completed, the material was dried and preserved.

### 2.4. Scale-Up Fermentation of the Wheat Straw

A scale-up experiment was conducted based on the previously optimized fermentation conditions. Fermentation samples were collected at 0, 5, 10, 15, and 20 days to analyze the fermentation performance of wheat straw, including the content of crude protein, true protein, and lignocellulose (lignin, cellulose, and hemicellulose).

### 2.5. Analysis Methods

Proteins (true and crude) were measured according to the Chinese national standard method [44]. Total DM was calculated based on the weight difference before and after drying in an oven at 105 °C. The concentrations of structural carbohydrates (i.e., cellulose and hemicellulose) and lignin were determined for each dried and ground sample using a two-step acid hydrolysis procedure based on the NREL method [44].

### 2.6. Statistical Analysis

All data were processed using Excel 2021 (Microsoft Co., Redmond, WA, USA) and visualized with Origin 2019b. Experimental data represent the mean ± standard deviation (SD) of three independent biological replicates (n = 3), where each replicate was prepared using separately inoculated fermentation substrates under identical conditions to ensure reproducibility. Statistical analysis was performed using IBM SPSS Statistics 25.0 (IBM Co., Armonk, NY, USA). The statistical analysis employed in this study was one-way analysis of variance (ANOVA). When the main effect reached statistical significance (*p* < 0.05), post hoc pairwise comparisons were conducted using appropriate methods based on data distribution characteristics.

## 3. Results and Discussion

### 3.1. Adaptive Laboratory Evolution of a Microbial Consortium Enhancing NPN Assimilation

To improve the conversion of NPN and wheat straw carbon, the microbial consortium collected from wheat rhizosphere soil was subjected to ALE to enhance NPN assimilation. Firstly, the growth of the microbial consortium was evaluated in a liquid medium containing varying concentrations of NPN sources (urea, ammonium sulfate, and ammonium chloride), with wheat straw serving as the sole carbon source. The results indicate that the microbial consortium exhibited limited growth (optical density at 600 nm below 0.4) at nitrogen concentrations exceeding 1 g/L (Appendix A). Optimal growth was observed at a nitrogen concentration of 0.1 g/L, where the nitrogen utilization efficiency for urea, ammonium chloride, and ammonium sulfate reached up to 93% (Appendix A). Thus, the microbial consortium was subjected to long-term evolutionary adaptation starting from 0.1 g/L nitrogen to enhance its NPN assimilation capabilities.

The microbial consortium was inoculated into a medium with wheat straw as the sole carbon source and NPN as the sole nitrogen source, with subculturing occurring at 72 h intervals. When the growth and NPN utilization capacity of the microbial consortium plateaued, the NPN concentration in the medium was increased stepwise. By the 20th generation, the microbial consortium was able to grow at an NPN concentration of 2 g/L, achieving an optical density at 600 nm (OD_600_) comparable to that observed under the initial nitrogen concentration conditions (Figure 1a–c). Although the nitrogen assimilation efficiency of the microbial consortium decreased with increasing NPN concentration, it gradually improved over successive generations when maintained at a constant nitrogen concentration (Figure 1d–f). At an NPN concentration of 2 g/L, the 20th microbial consortium exhibited nitrogen assimilation efficiencies of 17.45%, 26.01%, and 28.22% for urea, ammonium chloride, and ammonium sulfate, respectively. The corresponding nitrogen utilization amounts were 0.35, 0.52, and 0.56 g/L, all exceeding 0.1 g/L (Figure 1g–i).

### 3.2. NPN Assimilation Capacity of the Evolved Microbial Consortium

The growth of the evolved microbial consortium was further assessed in a liquid medium where wheat straw served as the sole carbon source, alongside varying concentrations of NPN, including urea, ammonium chloride, and ammonium sulfate. The results show the evolved microbial consortium exhibited an NPN tolerance up to 5 g/L, a four-fold increase than that of the original microbial consortium. This suggests that adaptive evolution significantly enhanced both NPN tolerance and assimilation capacity (Figure 2a–c). Within the NPN concentration range of 0–2 g/L, microbial growth progressively increased with rising nitrogen levels. However, when the NPN concentration exceeded 2 g/L, the microbial growth was inhibited.

Furthermore, the growth and nitrogen utilization of the original microbial consortium, along with the 5th, 8th, 15th, and 20th generations, were assessed in a liquid medium supplemented with 5 g/L of nitrogen and wheat straw as the sole carbon source. As shown in Figure 3a–c, the growth performance of the 5th-, 8th-, 15th-, and 20th-generation microbial consortia surpassed that of the original microbial consortium. The value of the OD_600_ of the 20th generation in medium containing ammonium sulfate reached 0.7, the highest among all generations with varying NPN levels. Consistently, the 20th generation in the medium with ammonium sulfate exhibited the highest biomass yield of 1648.78 mg/L (Figure 3d–f) and the highest nitrogen assimilation capacity (Figure 3g–i). These results suggest that the evolved microbial consortium in ammonium sulfate had the strongest NPN assimilation capacity, which would be selected for the feed protein production from ammonium sulfate and wheat straw.

### 3.3. Production of Feed Protein by Using the Original and the Evolved Microbial Consortium

To produce the feed protein from NPN (ammonium sulfate) and wheat straw, an aerobic SSF of wheat straw was conducted using both the original and evolved microbial consortium. As illustrated in Figure 4, the true protein content of the wheat straw increased following SSF with both microbial consortia. Specifically, after 30 days of SSF, the true protein content rose from 4.55% to 6.23% with the original microbial consortium and from 3.44% to 7.37% with the evolved consortium. Furthermore, the crude protein content of the wheat straw increased from 5.98% to 9.65% with the original microbial consortium and from 5.15% to 9.26% with the evolved microbial consortium, reflecting enhancements of 61.37% and 79.8%, respectively, relative to their initial levels (Appendix A). Indeed, the observed increase in true protein content is primarily attributed to the reduction in wheat straw mass and the conversion of NPN into protein. The similar increase in crude protein content indicated the weight loss was less differentiating for the wheat straw fermented by the original and evolved microbial consortium. However, the more substantial increase in true protein content in wheat straw fermented by the evolved microbial consortium, compared to the original microbial consortium, suggested that the evolved microbial consortium exhibited enhanced utilization of NPN (ammonium sulfate). During the SSF process involving wheat straw and ammonium sulfate, complex carbon macromolecules, such as cellulose, hemicellulose, and lignin in the wheat straw, were degraded into simpler compounds, like glucose, which facilitated microbial metabolism and contributed to weight loss [44,45]. Microbial proliferation necessitates a balanced metabolism of carbon and nitrogen. Thus, optimizing conditions, including the incorporation of NPN in the SSF process, is essential to further improve the true protein content of wheat straw.

### 3.4. Optimization of SSF Conditions to Improve the True Protein Content of the Wheat Straw

#### 3.4.1. The Effect of Alkali Pretreatment

Alkali pretreatment was applied to effectively disrupt the lignocellulosic structure of wheat straw, thereby enhancing its accessibility to microbial enzymes and facilitating more efficient protein synthesis during solid-state fermentation [46]. The wheat straw was pretreated with varying concentrations of ammonium solution (0%, 15%, 20%, and 28% *w*/*w*) and sodium hydroxide (0%, 2%, 4%, 6%, and 8% *w*/*w*). The true protein content of the SSF of the pretreated wheat straw by the evolved microbial consortium was subsequently compared. Among the treatment groups, the 15% ammonium hydroxide pretreatment yielded the highest true protein content at 8.75%, representing a 1.89% increase compared to the true protein content of the untreated control (0% ammonium hydroxide) (Figure 5a). Conversely, the NaOH pretreatment did not improve the true protein content relative to the control group. In all groups (Figure 5c), the true protein content exhibited a trend of an initial increase followed by a decrease. Moreover, all pretreatment groups exhibited a significant decrease in total lignocellulose content (Figure 5d). Consistently, the 15% ammonium hydroxide pretreatment demonstrated the greatest lignocellulose degradation ability (Figure 5b), which may be attributed to its strong delignification effect and ability to swell cellulose fibers, thereby increasing enzymatic accessibility and microbial colonization efficiency [14,47]. This enhanced degradation likely provided more fermentable substrates for microbial growth and protein synthesis during SSF. Thus, 15% ammonium hydroxide was selected to pretreat the wheat straw for subsequent SSF to produce protein.

#### 3.4.2. The Effect of Fermentation Time, Inoculation Amount, Nitrogen Source Addition Amount, Initial Moisture Content, Temperature, and Substrate Sterilization

The conditions of fermentation play a crucial role in influencing microbial growth and metabolic capacity. Systematic optimization of these parameters can facilitate the regulation and control of the fermentation process [48]. In this study, the fermentation parameters, including fermentation duration, inoculum amount, nitrogen source addition amount, initial moisture content, temperature, and substrate sterilization of the SSF, were optimized to improve the true protein content of wheat straw. As illustrated in Figure 6a, the true protein content of wheat straw initially increased over time, reaching a peak value of 8.87% after 10 days of fermentation. The maximum true protein content of 9.01% was observed with an inoculum size of 15%; however, exceeding this inoculum size resulted in a gradual decline in the true protein content (Figure 6b). Figure 6c demonstrates the impact of nitrogen source addition on the true protein content, with the highest content of 9.12% achieved at a nitrogen source addition level of 0.5%. Beyond this concentration, the true protein content of wheat straw progressively decreased. The true protein content of wheat straw was found to be at its highest, 9.81%, at a moisture content of 70% (Figure 6d). Figure 6e illustrates the effect of temperature on the SSF of wheat straw. An increase in temperature from 20 °C to 30 °C resulted in an elevation of the true protein content from 6.50% to 9.60%. However, temperatures exceeding 30 °C led to a reduction in the true protein content below 9.60%. Under sterilized conditions, the true protein content of wheat straw increased from 2.73% to 10.00%, whereas a comparable increase from 2.74% to 10.42% was observed under non-sterilized conditions (Figure 6f). Consistent with the previous studies, SSF utilizing non-sterilized substrate proved more effective than SSF using the sterilized substrate. Previous studies have similarly reported that non-sterilized conditions often outperform sterilized fermentation conditions in terms of nutrient enrichment [49]. In addition, non-sterilized treatments are cost-effective and suitable for large-scale production. Overall, these results demonstrate that the optimization of fermentation parameters can significantly improve the true protein content of wheat straw. The highest single-cell protein yield of 3.06 g/L was achieved when potato starch processing wastewater was fermented by *Candida utilis*, *Geotrichum candidum*, and *Candida tropicalis* under optimized fermentation conditions [50]. Taken together, the optimal conditions for the evolved microbial consortium included the use of ammoniated wheat straw at 15% (*w*/*w*), non-sterilization of the substrate, an inoculation volume of 15% (*v*/*w*), a nitrogen source addition amount of 0.5% (*w*/*w*), an initial moisture content of 70%, a fermentation temperature of 30 °C, and a fermentation duration of 10 days (Table 1).

### 3.5. Scale-Up of the SSF of the Wheat Straw to Produce Feed Protein

The SSF process for wheat straw was scaled up from 0.04 to 2.5 kg under the optimized conditions to assess the reliability of the evolved microbial consortium. As shown in Figure 7a, the true protein content of the wheat straw increased from 2.74% to 11.60%, while the crude protein content increased from 8.04% to 15.34% after 20 days of SSF. These values were comparable to the true protein content (10.42%) and crude protein content (15.01%) observed in the fermentation of 0.04 kg wheat straw. Notably, the highest true protein content was achieved after 20 days of SSF, differing from the optimized fermentation duration of 10 days, potentially due to the microbial consortium requiring an extended adaptation period. Although these results were lower than those achieved in the small-scale system following optimization, the trend was consistent. In the fermentation of 40 g, the optimal protein content was reached on day 10. However, in the scale-up experiment, protein accumulation continued beyond day 10, peaking on day 20 with a true protein content of 11.60%. In addition, after 20 days of fermentation, the content of cellulose, hemicellulose, and lignin in the wheat straw decreased to 12.26%, 1.86%, and 5.67%, respectively, corresponding to degradation rates of 58.69%, 63.60%, and 29.21% (Figure 7b). It was found that 82.4% of the ammonium salts was assimilated and converted into the microbial protein. And the content of the non-protein nitrogen in the fermented wheat straw was less than 0.1%, which is much lower than the additive amount of non-protein nitrogen in the feed for the bovine and pig [51]. Comparative studies with other microbial consortia further validate the efficacy of our approach. For instance, domesticated microbial consortium A_TMC1-3 produced only 6.3% protein from corn stover [39], while domesticated microbial consortium FA12 achieved 15% hemicellulose degradation in distiller’s grains [52]. This technology offers substantial economic potential through the utilization of low-cost agricultural residues and feasible unsterile fermentation conditions, which could significantly reduce production costs compared to traditional sterilized systems. In addition, this approach represents a significant advancement in waste management and sustainable protein production. By converting agricultural residues into high-value microbial protein, we simultaneously address two critical challenges: reducing organic waste accumulation and alleviating pressure on conventional protein supply chains. However, maintaining uniform conditions, such as temperature and nutrient supply, presents significant challenges during scale-up fermentation, as temperature gradients and uneven resource distribution can disrupt microbial community structure and function. Future research will systematically address these issues to guarantee the sustainability of fermentation.

In summary, these results highlight the scalability potential of the microbial consortium and the stability of the fermentation process, providing a robust foundation for future industrial applications.

## 4. Conclusions

This study focused on microbial consortia capable of degrading wheat straw and assimilating NPN. A microbial consortium with the capacity of efficient assimilation NPN through ALE was obtained. The nitrogen tolerance and utilization capabilities of the microbial consortium were evaluated. The SSF conditions for wheat straw using the evolved microbial consortium were optimized, including the use of ammoniated wheat straw (15%), non-sterilized substrate, 15% inoculation amount, 0.5% nitrogen addition amount, 70% initial moisture content, a fermentation temperature of 30 °C, and a fermentation period of 10 days. Under the optimal conditions, the true protein content of wheat straw increased from 2.74% to 10.42%. Additionally, the stability of SSF process of the microbial consortium for feed protein production was confirmed through the scaled-up fermentation of wheat straw. This study provides a sustainable approach to valorizing agricultural residues and NPN into high-protein feed, thereby enhancing the high-value utilization of agricultural and industrial wastes. It addresses both the scarcity of protein resources and the environmental challenges associated with conventional feed production.

## Figures and Tables

**Figure 1 microorganisms-13-01416-f001:**
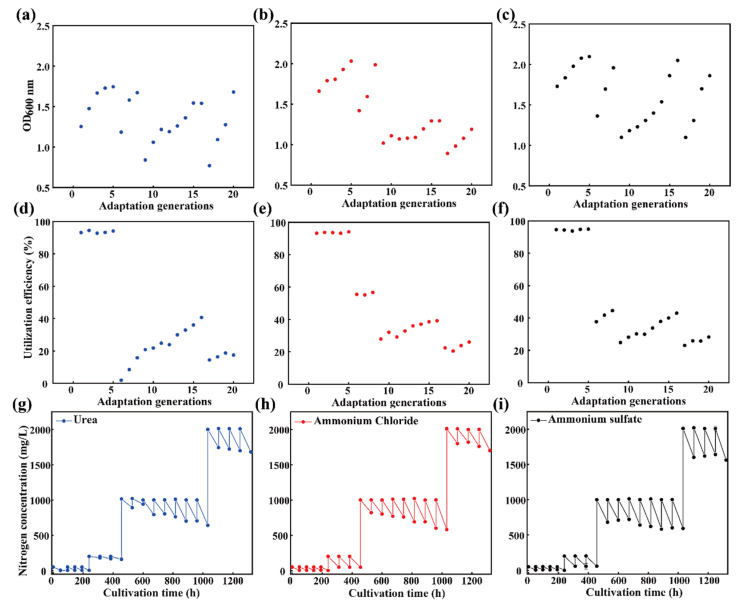
Adaptive laboratory evolution of the microbial consortium to enhance its NPN assimilation ability. The growth of the microbial consortium from different adaptation generations under different nitrogen sources (urea, ammonium chloride, and ammonium sulfate) (**a**–**c**). The nitrogen utilization efficiency in the medium for all the generations under different nitrogen sources (urea, ammonium chloride, and ammonium sulfate) (**d**–**f**). The nitrogen concentration in a medium for all the generations under different nitrogen sources (urea, ammonium chloride, and ammonium sulfate) (**g**–**i**).

**Figure 2 microorganisms-13-01416-f002:**
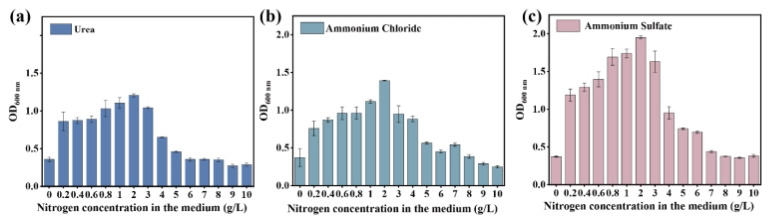
The growth of the evolved microbial consortium in medium with different concentrations of urea (**a**), ammonium chloride (**b**), and ammonium sulfate (**c**) as the sole nitrogen source and wheat straw as the sole carbon source. The data represent the averages of biological triplicates, and the scale bar represents the standard deviation (SD).

**Figure 3 microorganisms-13-01416-f003:**
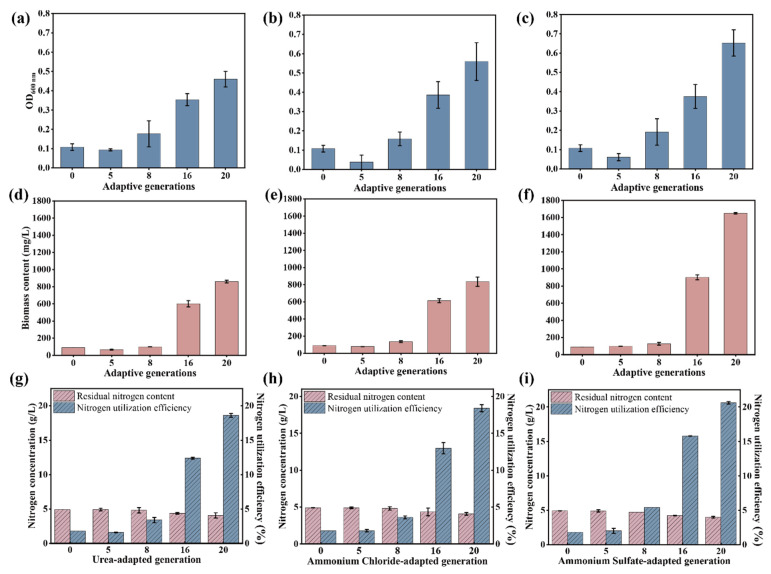
The NPN assimilation capacity of the 0th, 5th, 8th, 15th, and 20th generations of microbial consortium in liquid medium with NPN (5 g/L) as the sole nitrogen source and wheat straw as the sole carbon source for 72 h. The growth of the original and evolved microbial consortium under different nitrogen sources (urea, ammonium chloride, and ammonium sulfate) (**a**–**c**). The biomass yield of the original and evolved microbial consortium under different nitrogen sources (urea, ammonium chloride, and ammonium sulfate) (**d**–**f**). The nitrogen utilization of the original and evolved microbial consortium under different nitrogen sources (urea, ammonium chloride, and ammonium sulfate) (**g**–**i**). The data represent the averages of biological triplicates, and the scale bar represents the standard deviation (SD).

**Figure 4 microorganisms-13-01416-f004:**
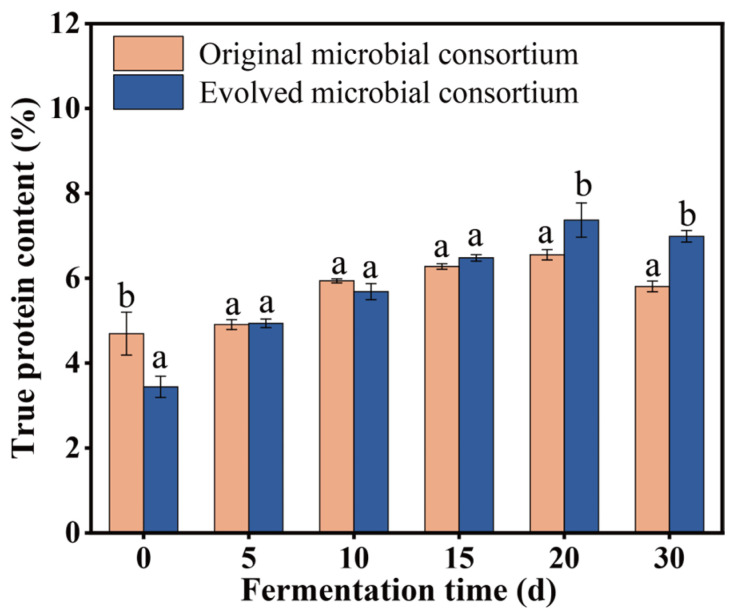
The true protein of the wheat straw fermented by using the original and the evolved microbial consortium with the addition of ammonium sulfate as a nitrogen source. The data represent the averages of biological triplicates, the scale bar represents the standard deviation (SD), and different lowercase letters indicate significant differences between treatments (*p* < 0.05).

**Figure 5 microorganisms-13-01416-f005:**
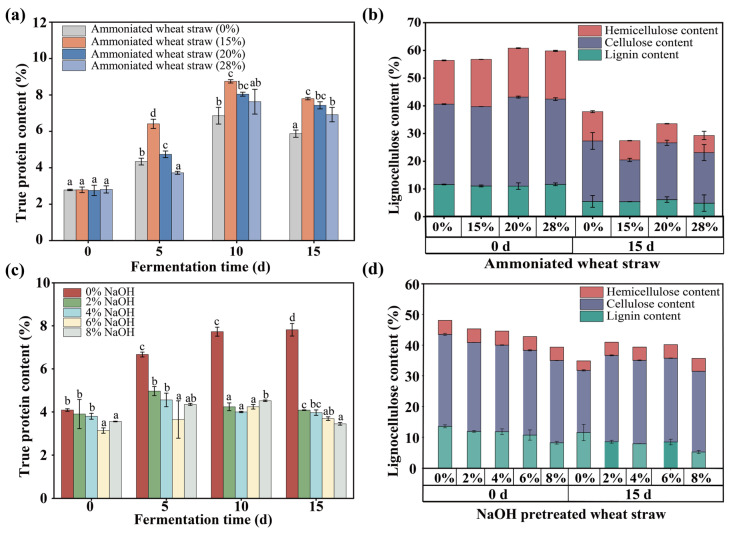
The changes in the true protein content and lignocellulose content (lignin, cellulose, and hemicellulose) during long-term SSF of wheat straw pretreated with different concentrations of ammonia and NaOH. True protein content (**a**,**c**); lignocellulose content (lignin, cellulose, and hemicellulose) (**b**,**d**). The data represent the averages of biological triplicates, the scale bar represents the standard deviation (SD), and different lowercase letters indicate significant differences between treatments (*p* < 0.05).

**Figure 6 microorganisms-13-01416-f006:**
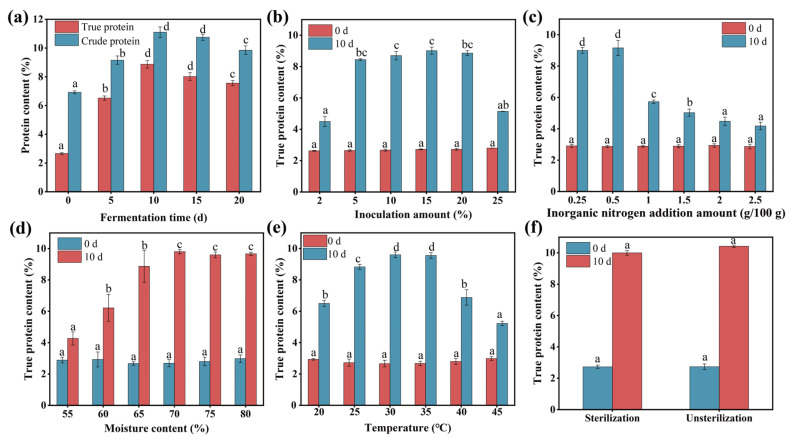
Protein content of SSF of ammoniated wheat straw by the evolved microbial consortium with different fermentation times (**a**), inoculation amounts (**b**), nitrogen source addition amounts (**c**), moisture contents (**d**), temperatures (**e**), and sterilized conditions (**f**). The data represent the averages of biological triplicates, the scale bar represents the standard deviation (SD), and different lowercase letters indicate significant differences between treatments (*p* < 0.05).

**Figure 7 microorganisms-13-01416-f007:**
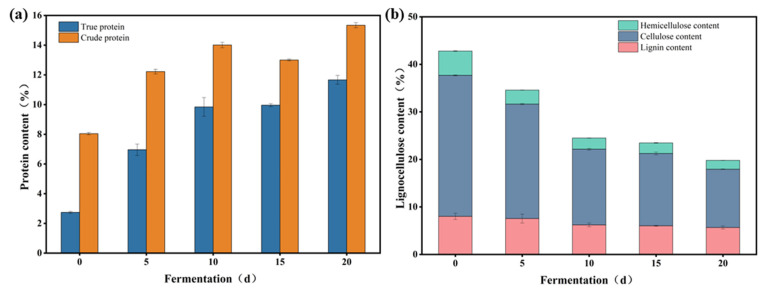
True protein and lignocellulose content of the wheat straw with the 2.5 kg SSF process. Protein content (**a**); lignocellulose content (lignin, cellulose, and hemicellulose) (**b**). The data represent the averages of biological triplicates, and the scale bar represents the standard deviation (SD).

**Table 1 microorganisms-13-01416-t001:** Optimized SSF conditions through single-factor experiments and corresponding true protein content of wheat straw.

Conditions	Pretreatment of Wheat Straw	Fermentation Time	Inoculation Amount	Nitrogen Addition Amount	Moisture Content	Temperature	Sterilization
Optimal conditions	15% ammonium hydroxide	10 days	15% *v*/*w*	0.5% *w*/*w*	70% *w*/*w*	30 °C	Non-sterilization
True protein content (%)		10.42 ± 0.07					

Note: Error values represent mean ± SD (n = 3 biological replicates).

## Data Availability

The original contributions presented in this study are included in the article/Appendix A. Further inquiries can be directed to the corresponding author.

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
