# Peer review of "Adaptive Laboratory Evolution of a Microbial Consortium Enhancing Non-Protein Nitrogen Assimilation for Feed Protein Production"

_microorganisms, 2025, doi:10.3390/microorganisms13061416_

Round 1

Reviewer 1 Report

Comments and Suggestions for Authors

The manuscript is interesting and belongs to the field of Biotechnology. It is important to find new sources and ways for animal feed production, mainly through microbial fermentation, which, in my opinion, is cheaper than other methods. the manuscript describes all the relevant data collected, supporting the initial hypothesis. Apart from some technical errors that have been included in the attached pdf, the authors should run response surface methodology (RSM) to provide the optimized parameters of fermentation. In addition, the authors must ensure how the addition of non-protein nitrogen could occur. By nitrogen salts used as food additives? Clarification is required for the application of this research in real-market analysis.

Based on the topic of interest addressed in this paper, I suggest a minor revision.

Comments on the Quality of English Language

The use of English language/words can be improved.

Author Response

Comments and Suggestions for Authors

Reviewer 1:

The manuscript is interesting and belongs to the field of Biotechnology. It is important to find new sources and ways for animal feed production, mainly through microbial fermentation, which, in my opinion, is cheaper than other methods. the manuscript describes all the relevant data collected, supporting the initial hypothesis.

Comment 1: Apart from some technical errors that have been included in the attached pdf, the authors should run response surface methodology (RSM) to provide the optimized parameters of fermentation.

Response: Thank you for your suggestions. In this manuscript, the true protein content of the wheat straw could reach 10.42% (Fig 6 in the file “Revised manuscript with marked”) through the single-factor tests of the various solid-state fermentation conditions, while the true protein content of the wheat straw could only reach 7.37% (Fig 4 in the file “Revised manuscript with marked”) with the unoptimized solid-state fermentation condition. Indeed, RSM has been applied for other solid-state fermentation in our lab, but the increase for the true protein content of the substrate is not much higher than the single factor test. For example, the true protein content of the Chinese distillers' grains could reach 19.01% with unoptimized solid-state fermentation conditions, and the true protein content was increased to 20.45% through RSM (Chen et al., International Journal of Food Microbiology, 2025, 435: 111154). Thus, the RSM was not performed to optimize the fermentation parameters in this manuscript.

Recently, the machine learning modeling has been employed to simulate the operation of the solid-state fermentation of the wheat straw using the microbial consortium. And the fermentation parameters with a higher true protein content have been predicted and verified, which will be published in our next article.

Comment 2: In addition, the authors must ensure how the addition of non-protein nitrogen could occur. By nitrogen salts used as food additives? Clarification is required for the application of this research in real-market analysis. Based on the topic of interest addressed in this paper, I suggest a minor revision.

Response: Thank you for your comments. The non-protein nitrogen, such as urea and ammonium salts, was recognized as a reliable nitrogen source for ruminants over 100 years ago, as the ruminant pre-gastric bacteria can utilize the non-protein nitrogen to produce microbial protein by digesting it in the small intestine, which becomes available to livestock (Niazifar et al., Heliyon, 2024, 10, e33752). It was also the feed supplemented with 1.5% ammonium phosphate could improve feed efficiency and increase lean depth of the pig (Miranda et al., Journal of Animal Science, 2023, 101, 285). In this manuscript, the non-protein nitrogen (ammonium salts) was not directly introduced into the feed, while the ammonium salts were used as the nitrogen source in the solid-state fermentation. The microbial consortium could metabolize NPN by hydrolyzing it into ammonia and incorporate it into microbial protein synthesis. In addition, this manuscript only added 0.5% non-protein nitrogen into the wheat straw for the solid-state fermentation and 82.4% of the non-protein nitrogen was converted into microbial protein (Table S2 in the file “Revised supplementary materials with marked”). The left content of non-protein nitrogen in the fermented wheat straw was less than 0.1%, which should be safety for the animals.

In the revised manuscript, the conversion rate of the ammonium salts in the solid-state fermentation has been added, and the clarification of the addition of the non-protein nitrogen was also added in lines 453-458 in the file “Revised manuscript with marked”.

Comment 3: These growth factors are chemicals in the nutrient-enriched medium. How can be substituted with natural growth factors?

Response: Thank you for your comment. The nutrient-enriched medium is a microbial seed culture medium specifically designed for cultivating the original microbial consortium. Currently, we use synthetic growth factors to ensure consistent and reproducible growth of the microbial community in the liquid seed culture system. However, we acknowledge that natural growth factors would be more desirable for ecological sustainability. In our future work, we plan to explore substituting these synthetic factors with natural alternatives while maintaining the stability and performance of the microbial consortium.

Reviewer 2 Report

Comments and Suggestions for Authors

Dear Authors,

The manuscript "Adaptive Laboratory Evolution of a Microbial Consortium Enhancing Non-Protein Nitrogen Assimilation for Feed Protein Production" presents a scientifically sound and timely study that addresses the global need for sustainable, non-soy-based animal feed alternatives. By applying adaptive laboratory evolution (ALE) to a microbial consortium and optimizing solid-state fermentation (SSF) of wheat straw and non-protein nitrogen (NPN), the authors offer a promising biotechnological route for enhancing feed protein production from agricultural waste. The study is well-designed, the results are clear, and the conclusions are supported by the data.
However, improvements in language clarity, figure presentation, and statistical reporting would significantly strengthen the manuscript’s quality and impact.

However, several critical aspects require attention to improve the manuscript's clarity, coherence, and overall impact:

Major Suggestions
1. Language and Grammar
The manuscript would benefit from a thorough English language revision to correct grammatical errors, awkward phrasing, and redundant wording (e.g., “plant plant-sourced” in the introduction), while also ensuring consistent verb tenses and maintaining technical accuracy throughout the text.

2. Statistical Analysis
Although data are presented as means ± standard deviation, the manuscript lacks statistical significance testing. To strengthen the validity of comparisons—such as between the original and evolved consortia or among different optimization conditions—please include p-values or other appropriate statistical indicators. This information should also be clearly incorporated into both the figure legends and the methods section.

3. Figures and Captions
Enhance the figure captions by providing more detailed experimental context, clearly defining all abbreviations, and specifying elements such as the content of each subpanel (a–i) and the meaning of error bars. Ensure that all figures are easily readable and appropriately spaced, particularly those containing multiple subplots. Additionally, include at least one summary table outlining the optimized SSF conditions alongside the corresponding protein yields to facilitate clearer interpretation and support the practical application of your results.

4. Methods Clarification
Clearly specify the exact NPN concentrations used at each stage of the ALE process, rather than only reporting the final value (5 g/L); a summary table illustrating the progression would greatly enhance clarity. Additionally, clarify whether biological replicates refer to independent cultures and indicate the number of replicates used for each experiment to ensure transparency and reproducibility.

5. Microbial Community Characterization
The study would be significantly enhanced by incorporating a basic microbial profiling analysis (e.g., 16S rRNA sequencing) to monitor shifts in community composition throughout the ALE process. If such analysis falls beyond the current scope, it is recommended that this be acknowledged and proposed as a direction for future research.

6. Discussion and Broader Impact
Consider expanding the discussion to address the economic feasibility or potential for techno-economic analysis of the process, as well as comparisons with related studies—particularly those involving SSF with evolved consortia. Additionally, discussing the environmental and regulatory implications of using microbial protein derived from agricultural waste would enhance the broader relevance of the work. It would also be beneficial to briefly acknowledge potential limitations, such as variability in microbial community performance during scale-up or under unsterile field conditions.

Minor Corrections
Abstract: “...which is 5-fold than that...” → “...which is fivefold higher than...”
Keywords: Avoid overly generic terms like "feed protein." Consider more specific alternatives such as “solid-state fermentation,” “adaptive laboratory evolution,” “ammonium sulfate assimilation.”

Conclusion
This is a well-executed and original study with clear practical relevance to sustainable feed production. With improvements in language clarity, statistical robustness, and the presentation of figures and tables, the manuscript will be strongly positioned for publication in a reputable journal.

Author Response

Reviewer 2

The manuscript “Adaptive Laboratory Evolution of a Microbial Consortium Enhancing Non-Protein Nitrogen Assimilation for Feed Protein Production” presents a scientifically sound and timely study that addresses the global need for sustainable, non-soy-based animal feed alternatives. By applying adaptive laboratory evolution (ALE) to a microbial consortium and optimizing solid-state fermentation (SSF) of wheat straw and non-protein nitrogen (NPN), the authors offer a promising biotechnological route for enhancing feed protein production from agricultural waste. The study is well-designed, the results are clear, and the conclusions are supported by the data. However, improvements in language clarity, figure presentation, and statistical reporting would significantly strengthen the manuscript’s quality and impact. However, several critical aspects require attention to improve the manuscript's clarity, coherence, and overall impact:

Comment 1: Language and Grammar

The manuscript would benefit from a thorough English language revision to correct grammatical errors, awkward phrasing, and redundant wording (e.g., “plant plant-sourced” in the introduction), while also ensuring consistent verb tenses and maintaining technical accuracy throughout the text.

Response: Thank you very much for your suggestions. We have corrected misspellings, grammatical errors, poor terminology, and redundant wording throughout the text. The revisions can be seen in lines 21-28, 31-40, 45-130, etc. in the revised manuscript named as “Revised manuscript with marked”.

Comment 2: Statistical Analysis

Although data are presented as means ± standard deviation, the manuscript lacks statistical significance testing. To strengthen the validity of comparisons-such as between the original and evolved consortia or among different optimization conditions-please include p-values or other appropriate statistical indicators. This information should also be clearly incorporated into both the figure legends and the methods section.

Response: Thanks for the comment. The p-values were analyzed for Fig. 4 (lines 354-358), Fig. 5 (lines 382-388), and Fig. 6 (lines 428-433) in the file “Revised manuscript with marked”. The statistical analysis method for the p-values was added in lines 241-249.

Comment 3: Figures and Captions

Enhance the figure captions by providing more detailed experimental context, clearly defining all abbreviations, and specifying elements such as the content of each subpanel (a-i) and the meaning of error bars. Ensure that all figures are easily readable and appropriately spaced, particularly those containing multiple subplots. Additionally, include at least one summary table outlining the optimized SSF conditions alongside the corresponding protein yields to facilitate clearer interpretation and support the practical application of your results.

Response: Thank you for your suggestions. All figure captions explicitly defined all abbreviations and subpanel contents. All the data represent the averages of biological triplicates, and the scale bar represents the standard deviation (SD), which has been indicated in each figure caption. The revisions can be seen in lines 282, 301, 318, 354, 382, 428, 475, in the revised manuscript named “Revised manuscript with marked”.

Table 1 has been added to summarize the optimized SSF conditions and corresponding protein yields to facilitate clearer interpretation and support the practical application of our results, in the revised manuscript named “Revised manuscript with marked” (Lines 435-436).

Comment 4: Methods Clarification

Clearly specify the exact NPN concentrations used at each stage of the ALE process, rather than only reporting the final value (5 g/L); a summary table illustrating the progression would greatly enhance clarity. Additionally, clarify whether biological replicates refer to independent cultures and indicate the number of replicates used for each experiment to ensure transparency and reproducibility.

Response: Thanks for your suggestions. The exact NPN concentrations used at each stage of the ALE process can be seen in Fig. 1 g, h, and i. Also, Table S1 was added to illustrate the concentrations of NPN (urea, ammonium chloride, and ammonium sulfate) at different generations during the ALE process in the file “Revised manuscript with marked”.

Except for the laboratory adaptive evolution process, all other experiments involving microbial consortium were conducted with biological replicates were performed with biological replicates comprising independently cultured samples. Specifically, three parallel cultures were established simultaneously under identical conditions in separate flasks to account for both biological variability and experimental consistency.

Comment 5: Microbial Community Characterization

The study would be significantly enhanced by incorporating a basic microbial profiling analysis (e.g., 16S rRNA sequencing) to monitor shifts in community composition throughout the ALE process. If such analysis falls beyond the current scope, it is recommended that this be acknowledged and proposed as a direction for future research.

Response: Thank you very much for your suggestion. As part of our ongoing research, we are conducting 16S rRNA gene sequencing analysis to analyze the shift in the microbial community throughout the ALE process. Moreover, a synthetic microbial consortium will be constructed with the help of machine learning, which will also be part content of our next article.

Comment 6: Discussion and Broader Impact

Consider expanding the discussion to address the economic feasibility or potential for techno-economic analysis of the process, as well as comparisons with related studies-particularly those involving SSF with evolved consortia. Additionally, discussing the environmental and regulatory implications of using microbial protein derived from agricultural waste would enhance the broader relevance of the work. It would also be beneficial to briefly acknowledge potential limitations, such as variability in microbial community performance during scale-up or under unsterile field conditions.

Response: Thank you for your suggestions. In this manuscript, the true protein content of the wheat straw could reach 11.60% (Fig. 7a in the file “Revised manuscript with marked”). The content of cellulose, hemicellulose, and lignin in the wheat straw decreased to 12.26%, 1.86%, and 5.67%, respectively, corresponding to degradation rates of 58.69%, 63.60%, and 29.21% (Fig. 7b in the file “Revised manuscript with marked”). Comparative studies with other microbial consortia further validate the efficacy of our approach. For instance, domesticated microbial consortium A_TMC1-3 produced only 6.3% protein from corn stover (Wang et al., Waste Management, 2025, 194: 298-308), while domesticated microbial consortium FA12 achieved 15% hemicellulose degradation in distiller’s grains (Zhang et al., Journal of Agricultural and Food Chemistry, 2024, 72(16): 9259-9267).

This technology offers substantial economic potential through the utilization of low-cost agricultural residues and feasible unsterile fermentation conditions, which could significantly reduce production costs compared to traditional sterilized systems. The process supports circular economy principles by transforming waste into valuable products, aligning with current regulatory trends favoring sustainable bioprocessing solutions. In addition, this approach represents a significant advancement in waste management and sustainable protein production. By converting agricultural residues into high-value microbial protein, we simultaneously address two critical challenges: reducing organic waste accumulation and alleviating pressure on conventional protein supply chains.

During scale-up fermentation, maintaining uniform conditions like temperature and nutrient supply becomes challenging, causing shifts in microbial community structure and function due to temperature gradients and uneven nutrient distribution in scale-up fermentation. Under unsterile field conditions, competing indigenous microbes can outcompete introduced ones for resources, while abiotic factors such as temperature fluctuations, humidity changes can significantly impact microbial growth, metabolic activity, and community performance. Recently, proteomic approaches have been employed to investigate microbial community succession during solid-state fermentation processes (Chen et al., Science of the Total Environment, 2024, 920: 171034). Consequently, comprehensive genomic analyses will be conducted to detect the changes in the microbial community before and after scaled-up fermentation, which will be published in our next article.

Comment 7: Minor Corrections

Abstract: “...which is 5-fold than that...” → “...which is fivefold higher than...”

Keywords: Avoid overly generic terms like “feed protein.” Consider more specific alternatives such as “solid-state fermentation,” “adaptive laboratory evolution,” “ammonium sulfate assimilation.”

Response: Thank you for the comments. We have changed “...which is 5-fold than that...” to “...which represents a fivefold increase compared to the original microbial consortium” in the Revised manuscript with marked.

The keywords have been updated to “Keywords: microbial consortium; adaptive laboratory evolution; wheat straw; non-protein nitrogen assimilation; solid-state fermentation”.

Reviewer 3 Report

Comments and Suggestions for Authors

Dear Authors,

I believe this manuscript aligns well with the aims and scope of the journal it was submitted to.

However, some improvements are needed before it can be considered for publication, as follows:

Line 23: "could tolerant" - not proper English grammar

Line 24: rephrase to "which is 5-fold that of..."

Line 42 and throughout the whole manuscript: in-text citation style does not conform to MDPI author guidelines (reference citation by numbers should be in-line in square brackets, not superscripts)

Line 46: the word "plant" is repeated

Line 58: there is no reference number 1415 in the reference list

Line 88: rephrase to "increasing by a factor of 1.05" or "increasing 1.05 times"

Line 101: there is no reference 374038 in the reference list; if these are references 37, 38 and 40 they should be listed in the proper order and with commas in between (i.e. "[37,48,40]" or "[37-38,40]")

Line 108: a round bracket closes, but when and where was it opened?

Line 128: How do you know this composition of wheat straws? Details of all analyses performed to obtain this data must be presented in the manuscript.

Line 130: "sterile saline" - indicate provider, vendor or manufacturer

Line 132: 1% is a concentration, not a ratio

Lines 144 & 180: for all these chemicals, please indicate their purities and vendors

Line 155: it is advisable to provide g-force in addition to rpms in the case of centrifuges. Also indicate details about the centrifuge (type, manufacturer, place, country)

Lines 160-161 and 193-197: Very poor scientific writing style. Passive voice verbs should be used.

Line 222: had grown (not "has", past perfect tense needed here)

Indicate details (type, manufacturer, place, country) for all major equipments used in all your experimental protocols. For example, what was the optical density measured with?

Line 300: comma or hyphen needed between 44 and 45

Figure 4 (and almost all other figures): On line 211 you claim to have used Excel and Origin for statistical testing, however no such statistics data are presented when making comparisons and drawing conclusions (nor any different letters marks or asterisk marks are shown on any of the charts to emphasize significant differences). Please add statistic testing results to validate your claims.

Line 398: the plural form is "consortia"

Reference List: what does the "[J]" symbol stand for? It is unusual, yet you present it when citing all references.

Comments on the Quality of English Language

The English is not the best and help from a professional English editor or native English speaker is much needed during revisions

Author Response

Reviewer 3

I believe this manuscript aligns well with the aims and scope of the journal it was submitted to. However, some improvements are needed before it can be considered for publication, as follows:

Comment 1: Line 23: "could tolerant" - not proper English grammar.

Response: Thank you for your suggestion. We have revised “could tolerant” to “demonstrated tolerance” in the revised manuscript with marked (Line 24).

Comment 2: Line 24: rephrase to "which is 5-fold that of...".

Response: Thank you for your suggestion. We have changed “which is 5-fold that of...” to “which represents a fivefold increase compared to the original microbial consortium” in the revised manuscript with marked (Line 26).

Comment 3: Line 42 and throughout the whole manuscript: in-text citation style does not conform to MDPI author guidelines (reference citation by numbers should be in-line in square brackets, not superscripts).

Response: Thank you for your suggestions. The reference formatting has been revised in its entirety in the latest revised manuscript with marked (numerical citations have all been placed in square brackets and are not superscripted).

Comment 4: Line 46: the word "plant" is repeated.
Response: Thank you for the reviewers' suggestion. The repeated word “plant” has been removed from the revised manuscript with marked (Line 49).

Comment 5: Line 58: there is no reference number 1415 in the reference list

Response: We appreciate the reviewers' question. The citation format for references 1415 was incorrect; it should have been presented as “14, 15”. The correction has been implemented in the revised manuscript with marked (Line 65).

Comment 6: Line 88: rephrase to "increasing by a factor of 1.05" or "increasing 1.05 times"

Response: Thank you for your detailed suggestions. The phrase “increasing by a factor of 1.05” has been revised to “a 1.05-fold increase” in the revised manuscript with marked (Line 98).

Comment 7: Line 101: there is no reference 374038 in the reference list; if these are references 37, 38 and 40 they should be listed in the proper order and with commas in between (i.e. "[37,48,40]" or "[37-38,40]")

Response: Thank you for the reviewers' suggestion. The citation format in the revised manuscript with marked (Line 112) has been corrected.

Comment 8: Line 108: a round bracket closes, but when and where was it opened?

Response: Thank you for your suggestion. The unnecessary parentheses have been removed from the revised manuscript with marked (Line 121).

Comment 9: Line 128: How do you know this composition of wheat straws? Details of all analyses performed to obtain this data must be presented in the manuscript.

Response: We sincerely appreciate the reviewers' constructive feedback. The compositional analysis of wheat straw was conducted in accordance with the methodology reported by Liu et al. (Bioresource Technology, 2023, 390: 129852). The details of this adaptation and its integration into our experimental procedures have been fully elaborated in the revised manuscript with marked.
line 232-237: Proteins (true and crude) were measured according to the Chinese national standard method[43]. Total DM was calculated based on the weight difference before and after drying in an oven at 105 ℃. The concentrations of structural carbohydrates (cellulose and hemicellulose, i.e.) and lignin were determined for each dried and ground sample using a two-step acid hydrolysis procedure based on the NREL method [43].

Comment 10: Line 130: "sterile saline" - indicate provider, vendor or manufacturer

Response: We appreciate the reviewers' suggestions and have supplemented the details of sterile saline in line 145 of the revised manuscript with marked.

Comment 11: Line 132: 1% is a concentration, not a ratio

Response: Thank you for your suggestion. We have revised the ambiguous descriptions of concentration and ratio, confirming that “1%” refers specifically to concentration (Line 148).

Comment 12: Lines 144 & 180: for all these chemicals, please indicate their purities and vendors

Response: Thank you for your suggestion. We have included the purity and supplier information for all chemicals in the revised manuscript with marked (Lines 151-157).

Comment 13: Line 155: it is advisable to provide g-force in addition to rpms in the case of centrifuges. Also indicate details about the centrifuge (type, manufacturer, place, country)

Response: Thank you for your suggestion. In the revised manuscript with marked, we have incorporated the centrifuge's operational parameters (specifically, the speed of 8000 rpm) and comprehensive equipment details into the experimental procedures, as clarified in Line 175.

Comment 14: Lines 160-161 and 193-197: Very poor scientific writing style. Passive voice verbs should be used.

Response: Thank you for your suggestion. In the revised manuscript with marked, we have rephrased the two sentences to address your concerns and incorporated the updated text into the manuscript.

Lines 184-188: 500 μL of the original microbial consortium bacterial solution was pipetted from the bacteria-preserving tube and inoculated into 100 mL of sterile nutrient-enriched medium. The culture was incubated at 30 ℃ in a shaking incubator at 150 rpm for 48 h, after which the activated bacterial suspension was obtained and used for subsequent experiments.

Lines 222-226: The dried wheat straw was crushed through a 40-mesh sieve (0.425 mm). Then, it was mixed evenly with NaOH solution at ratios of 1:1 (0%, 2%, 4%, 6%, and 8%). The mixture was placed into conical flasks, pressed tightly, and filled with a certain amount of CO₂. Subsequently, the flasks were incubated at room temperature for 10 days. After the treatment was completed, the material was dried and preserved.

Comment 15: Line 222: had grown (not "has", past perfect tense needed here)

Indicate details (type, manufacturer, place, country) for all major equipments used in all your experimental protocols. For example, what was the optical density measured with?

Response: Thank you for your suggestion. In the revised manuscript with marked, we have thoroughly revised the text to eliminate grammatical inaccuracies and supplemented the experimental procedures with detailed equipment specifications (e.g., Line 171-175: The growth of the bacterial culture was monitored by measuring the optical density (OD600 nm) of the bacterial solution at 600 nm using a UV–Vis spectrophotometer (Cary 60 UV–Vis, Agilent Technologies Inc., Santa Clara, CA, USA), and part of the bacterial solution was centrifugally separated by centrifuge (5810R, Eppendorf, China) at 8000 rpm (5356 x g) for 10 min to obtain the supernatant.).

Comment 16: Line 300: comma or hyphen needed between 44 and 45

Figure 4 (and almost all other figures): On line 211 you claim to have used Excel and Origin for statistical testing, however no such statistics data are presented when making comparisons and drawing conclusions (nor any different letters marks or asterisk marks are shown on any of the charts to emphasize significant differences). Please add statistic testing results to validate your claims.

Response: Thank you for your suggestion. In the revised manuscript with marked, we have performed rigorous statistical significance testing and annotated the figures (including Figure 4 (Lines 354-358), Figure 5 (Lines 382-387), and Figure 6 (Lines 428-433)) with letters denoting distinct significance levels.

Comment 17: Line 398: the plural form is "consortia"

Reference List: what does the "[J]" symbol stand for? It is unusual, yet you present it when citing all references.
Response: Thank you for your comment. The “[J]” marker in the references, used to indicate journal articles, was a citation error in the context of international academic writing. Following your recommendation, we have eliminated all “[J]” annotations from the Reference (Lines 518-624).

Round 2

Reviewer 3 Report

Comments and Suggestions for Authors

I am satisfied with the revisions